# Evaluation of Passive Strategies for Achieving Hygrothermal Comfort in Social Housing Buildings in the Dominican Republic

Dayana Acosta-Medina, Alberto Quintana-Gallardo * , Ignacio Guillén-Guillamón * and Fernando A. Mediguchia

Center for Physics Technologies, Universitat Politècnica de València, 46022 València, Spain;
acostadayana26@gmail.com (D.A.-M.); fermenfo@arq.upv.es (F.A.M.)
* Correspondence: alquigal@upv.es (A.Q.-G.); iguillen@upv.es (I.G.-G.)

**Abstract:** In a building, the thermal satisfaction an individual may experience generally influences their health, well-being, productivity, and energy consumption. The concept of thermal comfort and its importance in buildings has been known for some time. However, in the Dominican Republic, discussing thermal comfort in social housing is a challenge since there have not been many studies applied to this context, especially to social housing. For this reason, this research analyzed the thermal behavior of a typical social housing building through energy simulation, aiming to highlight the importance of passive strategies to improve comfort in a warm and humid climate without using air conditioning. The simulation was conducted using OpenStudio v3.9, which utilizes the EnergyPlus v9.4 calculation engine. Three case studies were analyzed, implementing passive measures and seeking to achieve temperatures within the comfort ranges of the housing prototype. The results show that combining different passive strategies for warm-humid climates significantly reduces temperature, achieving reductions of up to 2.8 °C in the colder period and up to 3.2 °C in the warmer period.

**Keywords:** thermal comfort; passive strategies; thermal simulation; social housing



## 1. Introduction

Humans have constantly sought to create and find thermally comfortable environments, as evidenced by traditional constructions worldwide. Unfavorable thermal conditions can lead to occupant dissatisfaction, which, in turn, negatively affects their productivity and performance [1]. Thermal comfort is defined, according to the American Society of Heating, Refrigerating, and Air-Conditioning Engineers (ASHRAE) [2], as "that mental condition that expresses satisfaction with the thermal environment." In other words, it represents the balance between a person's psychological and physiological sensations and their immediate surroundings [3]. This definition highlights the subjective nature of thermal comfort, which extends beyond simple temperature measurements.

The importance of thermal comfort in habitable spaces lies in its direct influence on people's health, well-being, and productivity, as well as on the energy consumption of buildings. A thermally comfortable environment is crucial for the optimal development of activities. Studies have shown that conditions outside the comfort zone, such as high temperatures or poor ventilation, can negatively impact health. Children, in particular, are more susceptible to unfavorable environmental conditions [4]. Determining thermal comfort standards is not simple and has led to different theoretical approaches.

The study of thermal comfort began in the early 20th century with the work of A. P. Gagge [5], who, in 1967, published the node model to explain the thermal balance of the

human body. Later, in 1973, P. O. Fanger [6,7] published his method for evaluating thermal comfort, which analyzes environmental and physiological parameters to determine thermal sensation using a scale of values. In 1962, R. K. Macpherson [8], in his publication on indoor environmental evaluation, noted that thermal comfort depends on physical variables such as air temperature, mean radiant temperature, air velocity, and relative humidity, as well as personal variables such as clothing insulation and activity level.

This approach seeks to establish universal comfort standards through mathematical models that relate environmental variables such as temperature, humidity, and air velocity to human thermal sensation. Neutral temperature, generally considered the comfort temperature, is determined through linear regression analysis, correlating subjective responses from field studies with objectively measured climatic parameters. Indexes such as the predicted mean vote (PMV) and the predicted percentage of dissatisfied (PPD), defined in international standards such as ISO 7730, are examples of this approach [9].

The qualitative approach, also known as the adaptive model, considers reality subjective and multifaceted. Studies based on this approach have shown that people accustomed to warm climates with high temperatures and humidity may have different thermal comfort ranges than those proposed by international standards [9]. A correlation has been found between neutral and average outdoor temperatures, particularly in naturally ventilated buildings [10].

Climate-adapted architecture emerges as a fundamental alternative for achieving thermal comfort passively, taking advantage of local climatic conditions and reducing the need for energy-intensive active climate control systems [4]. The climate-adapted design aims to integrate strategies such as natural ventilation [11], solar protection, materials with suitable thermal inertia, and favorable building orientation to optimize indoor comfort conditions [12].

In tropical countries, the energy consumption of cooling systems can account for up to 56% of total energy use [13], as the need for cooling indoor spaces is prevalent for most of the year. Passive cooling strategies for buildings offer ideal solutions for reducing this energy consumption [14]. In their study on passive envelope measures to improve energy efficiency in buildings in the United Arab Emirates, Friess and Rakhshan [15] highlight that passive design measures or retrofits represent one of the most effective strategies for creating energy-efficient building envelopes that minimize energy consumption without compromising human comfort.

In hot and humid climates, both temperature and humidity levels often exceed the permissible thresholds for human comfort, with high humidity posing the most significant challenge, as it hinders the effective implementation of passive systems to achieve comfortable conditions [16]. Consequently, using air conditioning and/or dehumidifiers has become increasingly common. The passive strategies that are demonstrated to be effective in such climates focus on reducing the transfer of thermal energy into the building and dissipating internal heat through ventilation [17].

Several authors emphasize that decision-making during the design process, particularly in its early stages, can improve comfort conditions and promote energy savings throughout the various phases of design [18,19]. Montoya Flórez and San Juan suggest that project decisions regarding appropriate orientation, proper ventilation, natural lighting, and selecting construction materials suited to the region and climate are key strategies that significantly contribute to achieving comfort while reducing reliance on mechanical cooling systems [12].

The Intergovernmental Panel on Climate Change (IPCC) has stated that global warming is unequivocal and directly linked to human activities. In this scenario, adapting to climate change and mitigating its effects by reducing energy consumption in the con-

struction sector have become priorities. Implementing passive and active climate control strategies based on alternative and low-greenhouse gas emission energy sources is becoming increasingly relevant [16]. Climate change disproportionately affects economically vulnerable populations, exacerbating existing inequalities and endangering their well-being. According to a United Nations report, over the past 20 years, 4.2 million people have been affected by natural disasters, and low-income countries have suffered estimated economic losses equivalent to 5% of their GDP due to these events [20].

In this context, selecting a social housing building for a study on thermal comfort and passive measures is particularly relevant. Social housing, which shelters vulnerable populations, often lacks efficient designs to mitigate extreme heat or cold, negatively impacting residents' health and quality of life. Implementing passive measures such as proper thermal insulation, natural ventilation, and sustainable construction materials can significantly improve thermal comfort and reduce dependence on active climate control systems, thereby lowering energy costs.

Several papers have studied passive strategies to mitigate thermal losses in social housing across Europe. This is the case of the study conducted by Gabriel Rojas and other researchers from the University of Innsbruck, who analyzed the thermal comfort of a social housing project in Austria. This social housing project, constructed according to the Passive House (PH) standard, showed some overheating problems during the summer, which highlighted the need for solar protection [21]. Similarly, an article published by Joe Forde and other researchers from the Building Energy Research Group of Loughborough University in the UK showed the risk of overheating in certain situations [22]. Outside of Europe, a study by Renata Dalbem, from the Federal University of Pelotas, Brazil, and other researchers showed that Passive House standards can lead to improving the thermal comfort of building users in the south of Brazil [23].

However, to our knowledge, the efficiency of simpler and more affordable passive strategies applied to social housing projects in the Dominican Republic has not yet been assessed. As explained in subsequent sections, social housing projects in the Dominican Republic lack any kind of climate adaptation strategy. Therefore, the primary objective of this study is to analyze how implementing passive methods influences the thermal comfort of a social housing building in the Dominican Republic. This research examines various passive cooling strategies that can improve comfort conditions in a hot tropical climate, aiming to meet the levels required by international standards without using mechanical ventilation. A typical social housing prototype located in the city of Santo Domingo is used as a case study for this investigation.

## 2. Materials and Methods

This study is structured as a case study comprising three phases: (1) Identification of the problem through formative and exploratory methodological techniques. (2) Selection of parameters and case study using analytical techniques and simulation. (3) Analysis of the results through confirmatory techniques. The main idea is to analyze the case study in four different situations. The first one should reflect the current situation of most social housing projects in Santo Domingo, which, as explained in the following sections, lacks any kind of climate adaptation strategy. The second situation includes the addition of solar protection. The third strategy adds utilizing natural ventilation strategies to avoid overheating. The last situation that must be studied is adding thermal insulation, which should be combined with the rest of the strategies. These four situations in which the case study is analyzed are depicted in Figure 1.

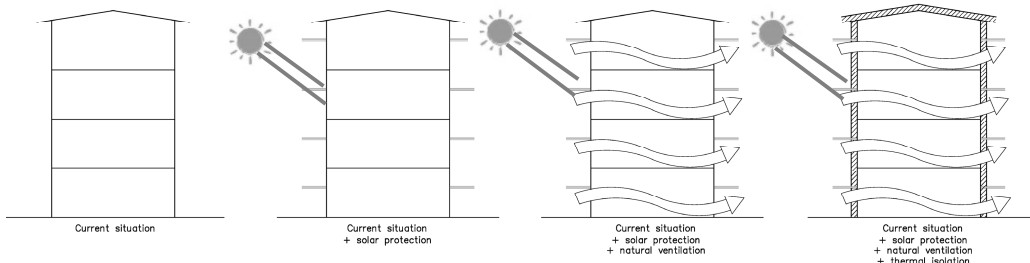

**Figure 1.** Passive strategies.

### 2.1. Case Study

The city of Santo Domingo is in the southern region of the Dominican Republic at latitude 18°29′08″ N and longitude 69°52′24″ W, with an elevation of 14 m above sea level. It has a hot tropical climate, with an average annual temperature of 25 °C. The average maximum temperature ranges between 29 °C and 32 °C, while the average minimum temperature varies between 19 °C and 21 °C. The city experiences a high annual rainfall of approximately 1447.1 mm, with the heaviest rainfall occurring from May to November, while winter months are drier. The high relative humidity, between 78% and 85%, characteristic of tropical climates, makes achieving adequate indoor comfort levels challenging.

Santo Domingo lies in the path of the northwestern trade winds, with an average annual wind speed of 10.1 km/h. Near the coast, wind direction shifts due to the temperature differential between land and water masses, causing winds to flow from sea to land during the day (SE) and from land to sea at night (NE). This phenomenon helps mitigate the constant heat and humidity conditions throughout the year. On average, Santo Domingo receives 215 h of sunshine per month, with 62% of the maximum possible insolation. The potential for global solar radiation in the Dominican Republic (average solar radiation on a horizontal surface) ranges from 5.25 to 5.50 kWh/m$^2$/day in the eastern half of the country and 5.50 to 6.00 kWh/m$^2$/day in the western half.

As a result, the thermal comfort issue in Santo Domingo buildings is primarily associated with the local climate, which remains consistently high throughout the year. Additionally, it is linked to the city's rapid growth and increasing population density, which leads to denser urban environments. In some cases, it is also related to construction methods and the quality of materials used in building these structures.

For several years, the government has undertaken numerous housing projects to address the existing housing deficit in the country, which, according to the National Statistics Office [24], increases by 3.13% annually. These projects are developed under low-cost housing policies, where material choices and the lack of application of sustainable construction standards have negatively impacted the thermal performance of the buildings.

The construction of this type of housing began in 1946 with the Social Improvement neighborhood, inaugurated by then-president Rafael L. Trujillo. These were small, isolated houses with tiny gardens. Later, in the 1970s and 1980s, during the presidency of Dr. Joaquín Balaguer, multifamily buildings with four levels were constructed, representing a completely different vision from what had been done previously, and this typology continues to be built to this day.

In the Dominican Republic, social housing programs focus on reducing the housing deficit quantitatively—by increasing the number of housing units—and qualitatively, aiming to improve the quality of existing homes [25]. These housing projects are primarily designed for low-income individuals, meaning that the household income must not exceed 40,000 Dominican pesos for families applying for a two-bedroom home and between 40,000 and 70,000 pesos for those applying for a three-bedroom home.

### 2.2. Case Study Selection

A typical residential building representing the standard construction model in the Dominican Republic was selected as the object of study. Figures 2 and 3 show the architectural floor plan of the building and the front elevation.

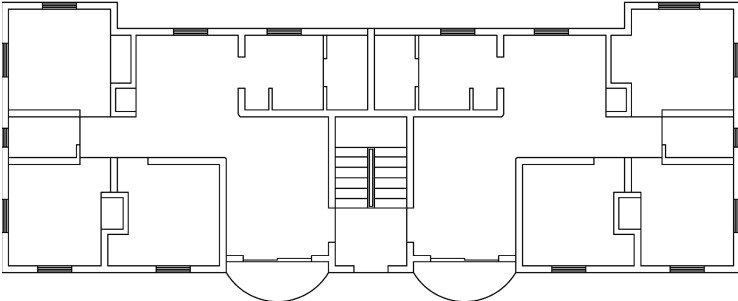

**Figure 2.** Architectural plan 1.

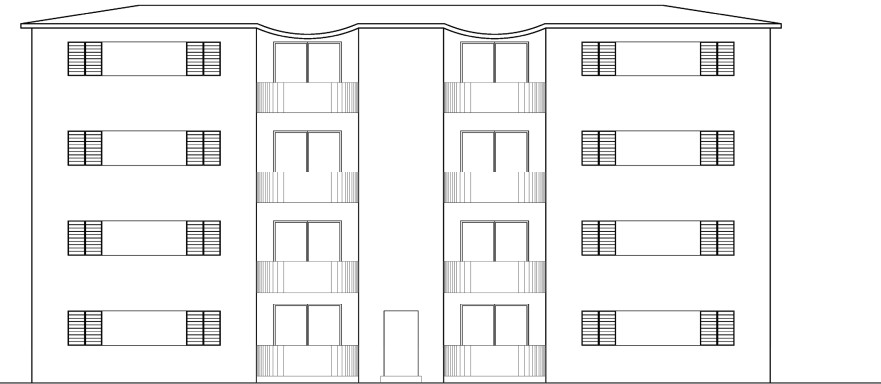

**Figure 3.** Architectural plan 2.

The building is oriented along a north–south axis. A thermal zone was designated for each level, and the ground floor, middle floor, and top floor were selected for the study. The study period was also defined, with two typical days chosen from different times of the year to enable a comparative analysis of the interior and exterior conditions at a given time. Temperature and relative humidity data were sourced from the American Society of Heating, Refrigerating, and Air-Conditioning and obtained from the OneBuilding.org platform in EPW format, with data recorded at 1-h intervals over 24 h.

To determine the thermal comfort index and the level of satisfaction, we used the static model and the adaptive model [19], both proposed by ANSI/ASHRAE Standard 55 [2]. The static model measures thermal sensation on a 7-point scale ranging from −3 to +3, with values between −0.5 and +0.5 considered the neutral comfort range. On the other hand, the adaptive model uses two levels of acceptability, 80% and 90%, to determine the comfort temperature range. For this study, the 80% comfort limit was used, calculated using the following formula:

$$\text{Upper limit of the comfort zone with 80\% acceptability} = 0.31 * T\_med + 21.3;$$
$$\text{Lower limit of the comfort zone with 80\% acceptability} = 0.31 * T\_med + 14.$$

where T_med refers to the average outdoor temperature [16].

The climatic variables used in both methods include air temperature, mean radiant temperature, and average outdoor temperature, which were taken from the climate data file. The air velocity was set at an average value between 0.1 m/s and 0.3 m/s. For the physiological variables, average values of 0.5 Clo for clothing insulation and 1.2 Met for

metabolic rate were used. The clothing insulation performance is considered the same for the cool and the hot period. The reason for this is the low-temperature differences during the year. This has been confirmed through informal interviews with residents of Santo Domingo. The floors included in the study are the first (N1), second (N2), and fourth (N4). The third floor was not included because it can be considered equivalent to N2, as neither is in contact with the floor or the roof.

*2.3. Simulation*

In this phase, thermal simulations were conducted. Due to the difficulty of taking and using expensive measuring equipment in the area, the paper is based purely on simulations. In situ measurements are outside the scope of this study. First, a reference model was created by simulating the current conditions of the case study. EnergyPlus™ software v9.4 and the OpenStudio graphical interface were used to model the building. Table 1 describes the composition and thermal properties of the materials used in the facade. According to data from the National Household Survey of Multiple Uses conducted by the National Statistics Office, the primary materials used in the country include concrete blocks for external walls and partitions, zinc and concrete for the roof, and aluminum for the windows.

**Table 1.** Elements of the façade construction materials in the Dominican Republic.

| Building Elements | Thermal Resistance m$^2$-$^{\circ}$K/W | Thermal Conductivity W/m $^{\circ}$K | Value (U) W/m$^2$ $^{\circ}$K |
|---|---|---|---|
| Cement mortar | 0.017 | 1.30 | 5.39 |
| Façade concrete block | 0.13 | 1.18 | 3.37 |
| Roof concrete | 0.13 | 1.65 | 3.30 |
| Aluminum windows | / | 230 | 5.84 |
| Wood doors | 250 | 0.16 | 0.37 |

The building features a flat roof with a 12 cm reinforced concrete slab. Its exterior surface is coated with a layer of mortar and a waterproofing membrane. The exterior walls and partitions are constructed using 20 cm thick concrete blocks, covered on both sides with a layer of cement mortar. The floor consists of a reinforced concrete slab, with horizontal aluminum sheets spaced ten centimeters apart across all surfaces. The entrance door is made of pine wood, while the interior walls are composed of plywood. Table 2 details the composition of the roof layers.

**Table 2.** Roof element construction materials of the Dominican Republic.

| Layer | Material | Thickness (mm) | Thermal Resistance m$^2$-$^{\circ}$K/W | Value (U) W/m$^2$ $^{\circ}$K |
|---|---|---|---|---|
| 1 | Waterproofing | 3 | | |
| 2 | Rustic mortar | 50 | | |
| 3 | Reinforced concrete | 120 | 0.29 | 3.48 |
| 4 | Rustic mortar | 20 | | |
| 5 | Rubbed mortar | 5 | | |

To determine the thermal comfort needs and potential strategies, established methods were employed, such as the Givoni psychrometric chart, published in 1969 in his book *Man, Climate and Architecture* [26]; the parameters proposed by Viktor Olgyay in his work *Design with Climate* [27]; and the adaptive comfort model parameters from ASHRAE Standard

55-2010 [2]. Both Givoni and Olgyay agree that two primary strategies should be applied in hot and humid climates: solar protection and natural ventilation. They also indicate that passive strategies may be insufficient in hot climates.

Figure 4 shows the psychrometric chart for the city of Santo Domingo, based on climate file data and generated using Climate Consultant software v6.0. The chart indicates that there are only 9 h of comfort without applying any strategy, representing 0.1% of the year. However, with the recommended strategies, up to 8760 h of comfort, or 100%, can be achieved. Natural ventilation and solar protection are the measures that most contribute to increased comfort hours.

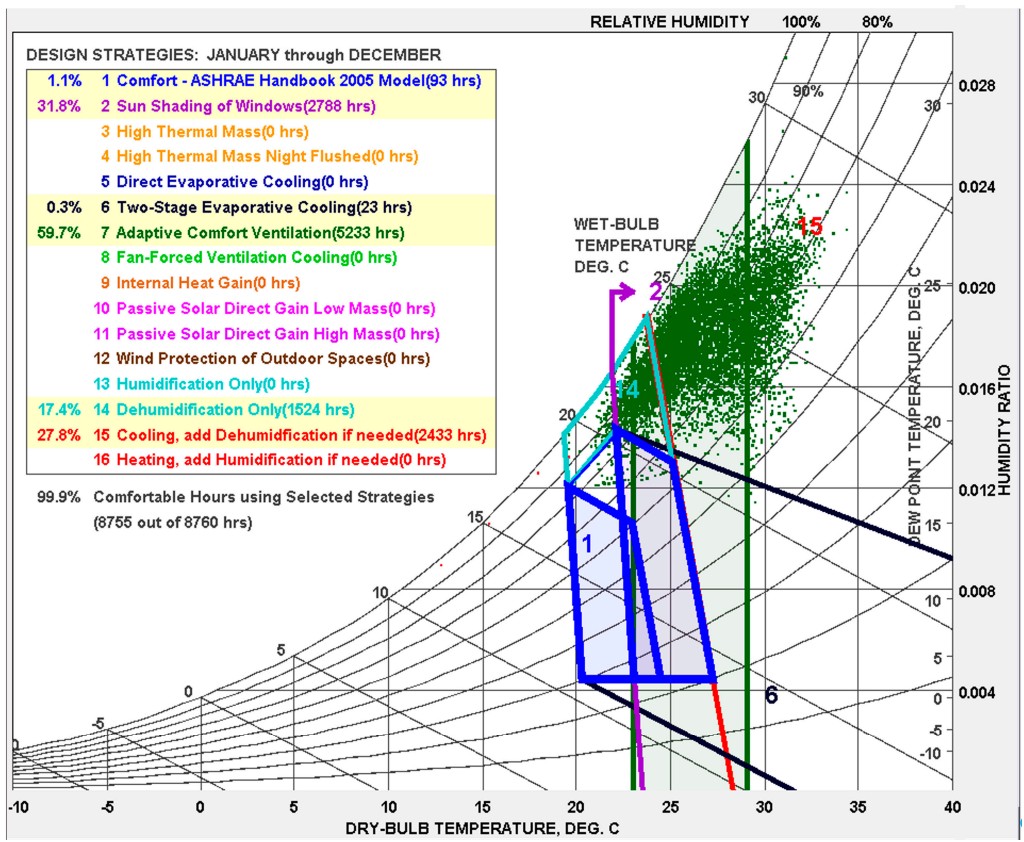

**Figure 4.** Givoni psychometric comfort strategies chart.

The strategies for this type of building must be as cost-effective as possible since it involves social housing. Therefore, Givoni's diagram and the studied literature selected the most affordable options: (1) Solar protection for windows, (2) allowing nighttime natural ventilation, and (3) insulating the facade and roof with 6 cm thick rock wool. Three scenarios were defined as follows:

1. The base model + solar protection;
2. The base model + solar protection + natural ventilation;
3. The base model + solar protection + natural ventilation + insulation.

These scenarios aim to maximize comfort while remaining affordable.

## 3. Results

### 3.1. Baseline Simulation: Case 0

The simulation results for the selected cool day and the baseline case (∅) are presented. The minimum outdoor temperature ($T_{ext}$) was 21.6 °C, and the maximum was 25.2 °C. In contrast, the building's average indoor temperature ($T_{int}$) ranged from a minimum of

27.6 °C to a maximum of 29.4 °C. These results reveal the temperature fluctuations between the minimum and maximum exterior temperatures, where the external temperatures are lower, showing a wide variation of 3.6 °C during this period, while the internal temperatures are higher, with a minor fluctuation of only 1.8 °C. The difference between the exterior and interior temperatures demonstrates the building's poor thermal performance in moderating these temperature variations, which, as mentioned earlier, is primarily due to the local climatic conditions.

Additionally, the average mean radiant temperature (Tmrad) ranged from 28.1 °C to 29.4 °C, fluctuating similarly to the indoor temperature. This further confirms the building's low thermal performance, as it cannot adequately regulate the interior thermal environment. Figure 5 shows the base case scenario's average interior and exterior temperature.

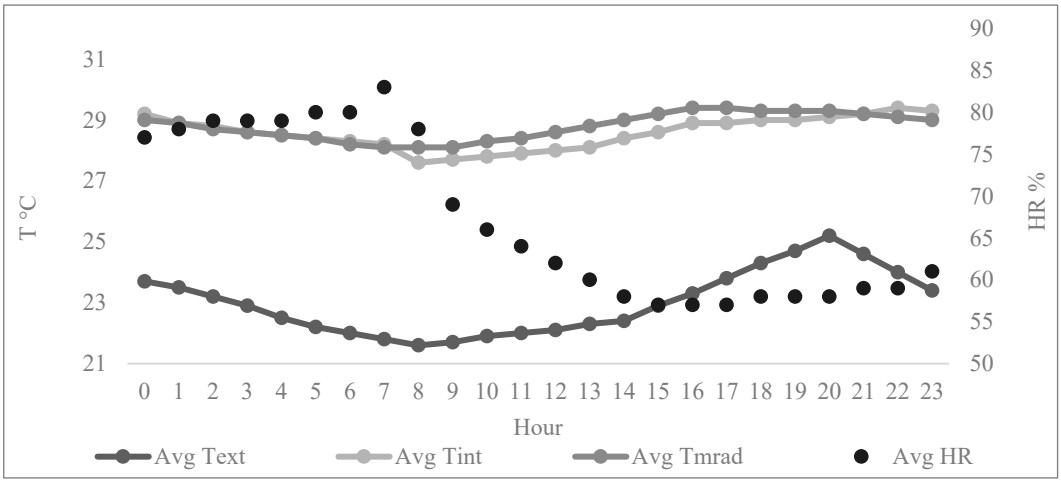

**Figure 5.** Average temperatures and relative humidity of the building for case Ø on the cool day.

In more detail, in Table 3 it was also observed that the worst thermal comfort conditions were recorded on the top floor (level 4), where the minimum temperature reached 28.4 °C and the maximum 31.4 °C, with an average relative humidity of 61%. In contrast, the ground floor (level 1) exhibited the best performance, with indoor temperatures (Tint) ranging between 23.9 °C and 25.1 °C and a relative humidity of 87%. In both cases, the mean radiant temperature fluctuated similarly to the indoor temperature, which can be attributed to significant thermal losses through the roof and floor. On the middle floor (level 2), the Tint ranged from 28.3 °C to 30.0 °C, with a relative humidity of 63%. These differences highlight the thermal challenges at various building levels due to heat gain and loss.

**Table 3.** Temperature range during the cool period for case Ø.

| Environmental Variables | 1st Level (N1) | 2nd Level (N2) | 4th Level (N4) |
|---|---|---|---|
| Avg outside temperature | | 20.8–27.7 °C | |
| Avg Indoor temperature | 23.9–25.1 °C | 28.3–30 °C | 28.4–31.4 °C |
| Avg mean rad temperature | 23.8–24.9 °C | 29.0–29.7 °C | 29.0–32.1 °C |
| Avg relative humidity | 76–98% | 53–80% | 48–80% |

The simulation results for the selected warm day and the baseline case (Ø) are presented (Figure 6 and Table 4). The minimum outdoor temperature (Text) was 27.8 °C, and the maximum reached 31.4 °C. Meanwhile, the building's average indoor temperature (Tint) ranged between 30.4 °C and 32.9 °C. On a cool day, the external temperature showed a wide fluctuation of 3.6 °C, whereas the indoor temperature fluctuated only by 2.5 °C. This indicates that the thermal performance of the building remains poor during this period.

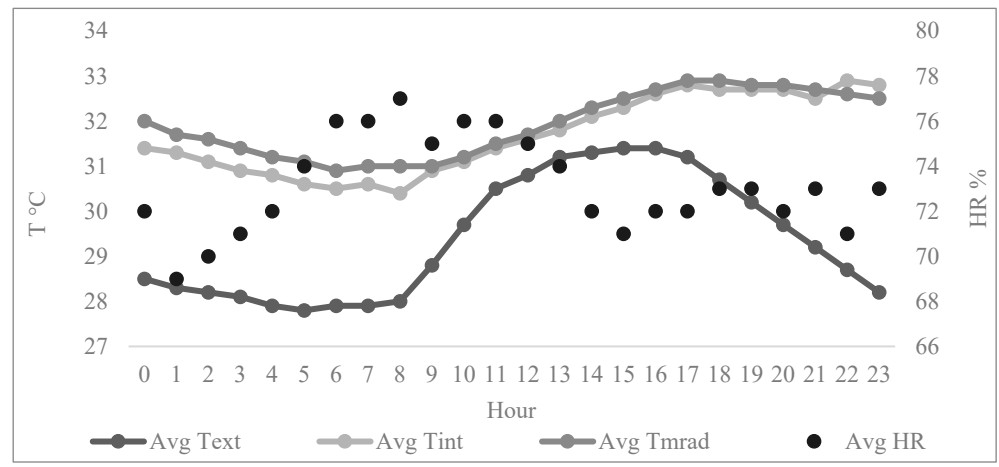

**Figure 6.** Average temperatures and relative humidity of the building for case Ø on the hot day.

**Table 4.** Temperature range during the hot period for case Ø.

| Environmental Variables | 1st Level (N1) | 2nd Level (N2) | 4th Level (N4) |
|---|---|---|---|
| Avg outside temperature | | 26–32.5 °C | |
| Avg Indoor temperature | 25.8–26.7 °C | 30.7–32.6 °C | 32.6–38 °C |
| Avg mean rad temperature | 25.5–26.3 °C | 31.2–32.3 °C | 33.6–38.7 °C |
| Avg relative humidity | 85–100% | 69–76% | 53–69% |

The graph illustrates that Tint is higher during the afternoon and evening, precisely when Text is lower, further demonstrating the building's inability to regulate internal heat effectively. The mean radiant temperature (Tmrad) ranged between 30.9 °C and 32.9 °C, closely mirroring the indoor temperature, confirming the building's insufficient thermal performance in warmer conditions.

As with the cool period, Table 5, the worst thermal comfort conditions during the warm period were also observed on the top floor (level 4), with minimum temperatures of 32.6 °C and maximum temperatures of 38 °C, along with an average relative humidity of 60%. On the ground floor (level 1), the indoor temperature (Tint) ranged from 25.8 °C to 26.7 °C, with a relative humidity of 96%. In both cases, the mean radiant temperature (Tmrad) fluctuated similarly to the indoor temperature. On the middle floor (level 2), Tint ranged between 30.7 °C and 32.6 °C, with a relative humidity of 71% and a Tmrad fluctuating between 30.9 °C and 32.9 °C.

**Table 5.** PMV for the cool period for case Ø.

| Variable | 1st Level (N1) | 2nd Level (N2) | 4th Level (N4) |
|---|---|---|---|
| Avg PMV | −0.1 | 1.4 | 1.6 |
| Avg PPD % | 11% | 42% | 58% |

Using the static model to calculate the comfort index, the predicted mean vote (PMV) for level 1 was −0.1, indicating a neutral thermal sensation. For level 2, the PMV was 1.4, and for level 4, it reached 1.6, indicating a slightly warm environment. These values reflect the thermal challenges at higher levels, particularly on the top floor, where discomfort is more pronounced.

For the warm day, Table 6, the average PMV value was 0.4, indicating a neutral thermal sensation. For level 2, the PMV was 2.2, indicating a warm environment, while for level 4, the PMV reached 3.5, signifying a hot thermal sensation.

**Table 6.** PMV for the hot period for case Ø.

| Variable | 1st Level (N1) | 2nd Level (N2) | 4th Level (N4) |
|---|---|---|---|
| Avg PMV | 0.4 | 2.2 | 3.0 |
| Avg PPD % | 19% | 82% | 99% |

Applying the equation from the adaptive model, the comfort range was determined to be between 22.6 °C and 29.6 °C. Using this model, during the cool day (Figure 7), the indoor temperature (Tint) at levels 2 and 4 was outside the comfort range for approximately 92% of the hours, with only 8% of the hours falling within the comfort range, mainly in the early morning when the outdoor temperature was very low. On the other hand, level 1 remained within the comfort range 100% of the time, both on cool and warm days. On the warm day, levels 2 and 4 were uncomfortable for 100% of the hours (Figure 8).

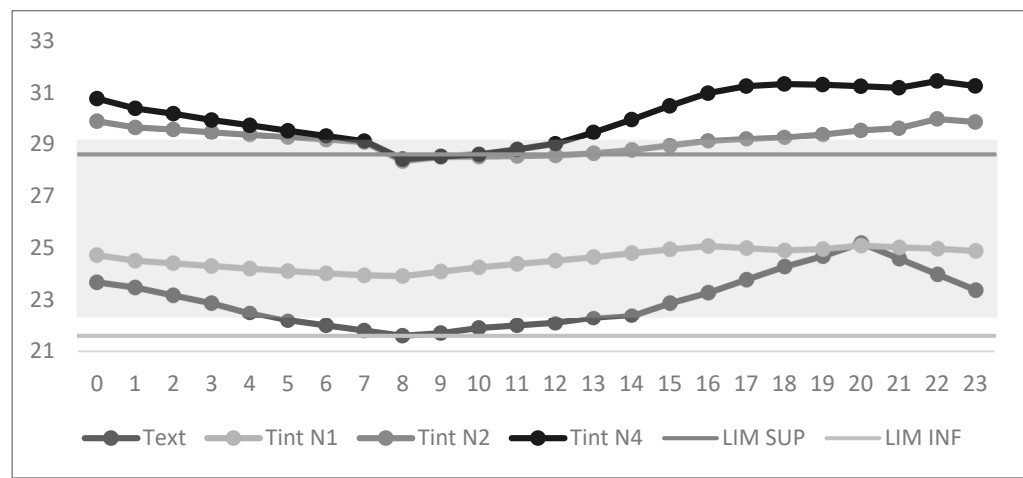

**Figure 7.** Outdoor and indoor temperatures on the cool day for case Ø.

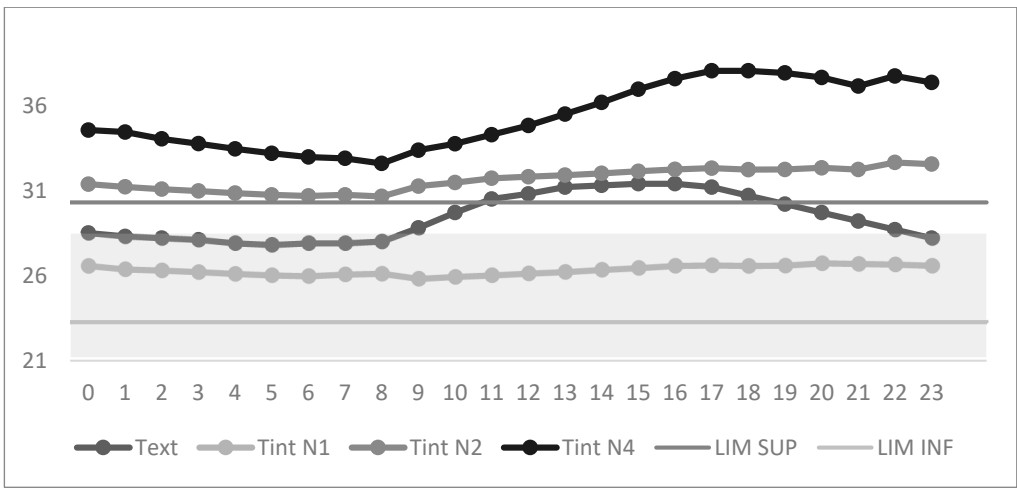

**Figure 8.** Outdoor and indoor temperatures on the hot day for case Ø.

Both models show that the intermediate level (N2) and the upper level (N4) are outside the comfort zone during both periods studied, while level 1 consistently demonstrates the best comfort conditions.

*3.2. Applying Passive Strategies*

3.2.1. Case 1: Solar Protection

The results showed that for the cool day in case 1, the building's indoor temperature ($T_{int}$) ranged between 27.1 °C and 28.6 °C, with a fluctuation of 1.5 °C (Figure 9). The mean radiant temperature ($T_{mrad}$) ranged between 27.1 °C and 28.3 °C, fluctuating similarly to the $T_{int}$. A general reduction in temperature compared to the baseline case (Ø) was observed, with $T_{int}$ decreasing by 0.8 °C and $T_{mrad}$ by 1.1 °C.

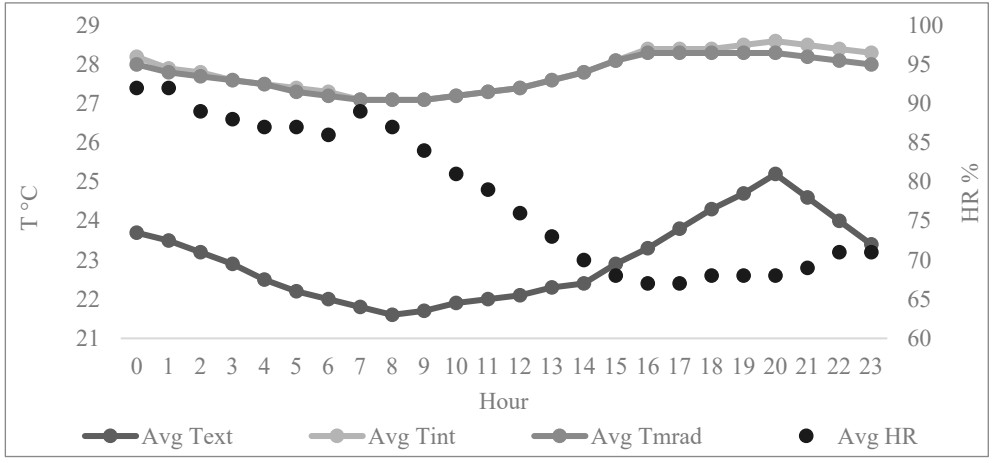

**Figure 9.** Average temperatures and relative humidity of the building on the cool day for case 1.

By applying sun protection using awnings, Table 7 shows that internal temperature fluctuations decrease compared to case 0. Similarly, the maximum internal temperature decreases, with the middle floor (level 2) showing the most significant reduction with a decrease of up to 1.2 °C, followed by the ground floor (level 1) with a reduction of 0.7 °C. The upper floor (level 4) showed the smallest reduction, with a decrease of 0.1 °C. The $T_{mrad}$ followed a similar pattern, with reductions of 0.9 °C at level 1, 1.3 °C at level 2, and 1.1 °C at level 4. This is likely due to the predominantly vertical path of the sun, meaning that the roof receives the most significant amount of radiation.

**Table 7.** Temperature range during the cool period for case 1.

| Environmental Variables | 1st Level (N1) | 2nd Level (N2) | 4th Level (N4) |
|---|---|---|---|
| Avg outside temperature | | 20.8–27.7 °C | |
| Avg Indoor temperature | 23.3–24.4 °C | 27.7–28.8 °C | 28.1–31.3 °C |
| Avg mean rad temperature | 23.2–24.0 °C | 27.8–28.4 °C | 28.1–31.3 °C |
| Avg relative humidity | 79–99% | 66–92% | 59–89% |

For the warm day (Figure 10), the building's $T_{int}$ ranged between 29.9 °C and 32.4 °C, with a fluctuation of 2.5 °C and relative humidity levels of 78%. The $T_{mrad}$ ranged between 30.2 °C and 32.1 °C. A slight overall decrease was observed, with $T_{int}$ dropping by 0.5 °C and $T_{mrad}$ by 0.7 °C.

As in the cold day scenario, internal temperature fluctuations did not show significant changes, as shown in Table 8. The middle floor (level 2) performed best, with a 1 °C reduction in $T_{int}$ and a 1.1 °C reduction in $T_{mrad}$, followed by the lower floor (level 1), where Ink decreased by 0.5 °C and $T_{mrad}$ by 0.5 °C. The upper floor (level 4) showed a slight increase of just −0.1 °C in Ink and 0.6 °C in $T_{mrad}$.

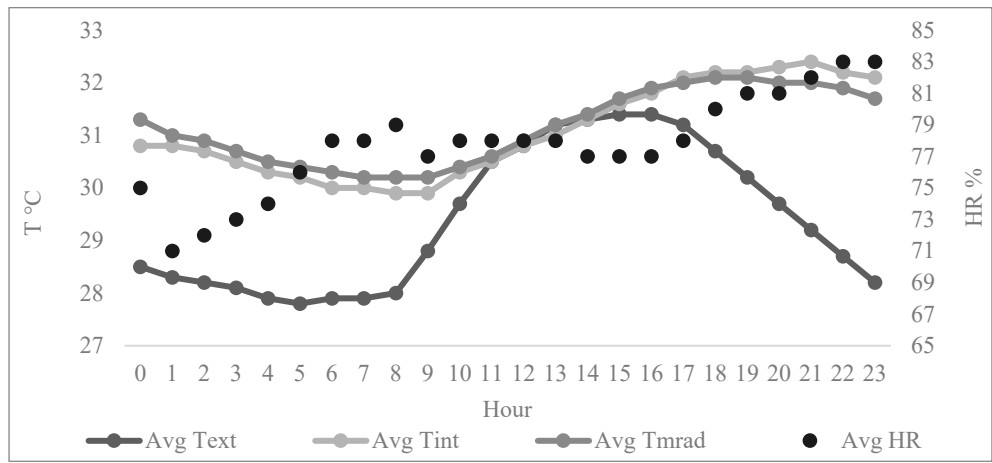

**Figure 10.** Average temperatures and relative humidity of the building on the hot day for case 1.

**Table 8.** Temperature range during the hot period for case 1.

| Environmental Variables | 1st Level (N1) | 2nd Level (N2) | 4th Level (N4) |
|---|---|---|---|
| Avg outside temperature | | 26–32.5 °C | |
| Avg Indoor temperature | 25.3–26.3 °C | 29.9–31.6 °C | 32.2–38.1 °C |
| Avg mean rad temperature | 25.1–25.8 °C | 30.3–31.2 °C | 32.9–38.1 °C |
| Avg relative humidity | 90–100% | 72–86% | 58–70% |

The PMV values indicate a neutral thermal sensation on level 1, with a value of −0.4 (Table 9). For level 2, the PMV was 1, and for level 4, it was 1.4, indicating a slightly warm environment. These results underscore the impact of solar protection measures, particularly in reducing discomfort on the middle and lower levels of the building.

**Table 9.** PMV for cool and hot periods for case 1.

| | Cold Period | | | Hot Period | | |
|---|---|---|---|---|---|---|
| | 1st Level (N1) | 2nd Level (N2) | 4th Level (N4) | 1st Level (N1) | 2nd Level (N2) | 4th Level (N4) |
| Avg PMV | −0.4 | 1 | 1.4 | 0.4 | 2 | 3.4 |
| Avg PPD % | 13% | 27.1% | 47.6% | 16% | 73% | 98% |

For the warm day, the average PMV was 0.4, indicating a neutral thermal sensation (Table 9). For level 2, the PMV was 2, reflecting a warm environment, while for level 4, it was 3.4, indicating a hot thermal sensation. The reduction in PMV was minimal on both days. Although there was a slight decrease in Tint and Tmrad, the high humidity levels prevented any significant improvement in thermal sensation.

When comparing the thermal performance results of the solar protection strategy applied to the base case with the comfort range from the adaptive model, we observe that, on a cool day, the Tint on the middle level (N2) remained within the comfort range for 100% of the hours, achieving a 92% increase in comfortable hours compared to the base case (Figure 11). On the top level (N4), 30% of the hours were within the comfort range, gaining 21% compared to the base case, with the afternoon and evening hours falling outside the comfort range. On both the cool and warm days, level 1 remained within the comfort range throughout the day, as in the previous case.

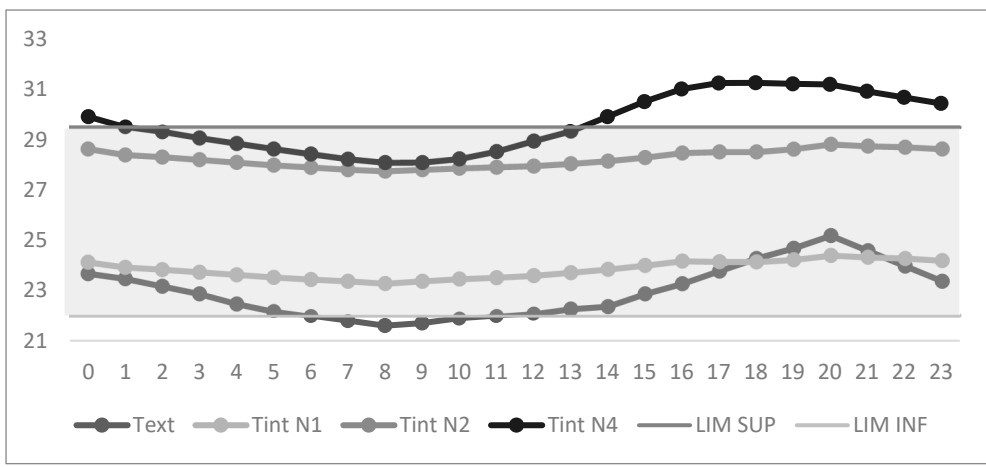

**Figure 11.** Outdoor and indoor temperatures on the cool day for case 1.

For the warm day (Figure 12), the middle level (N2) achieved 12% of the hours in comfort compared to the base case (Ø), while the top level (N4) remained in discomfort for 100% of the hours. Despite applying solar protection strategies, these results highlight the challenges of achieving thermal comfort on higher floors, particularly under warm conditions.

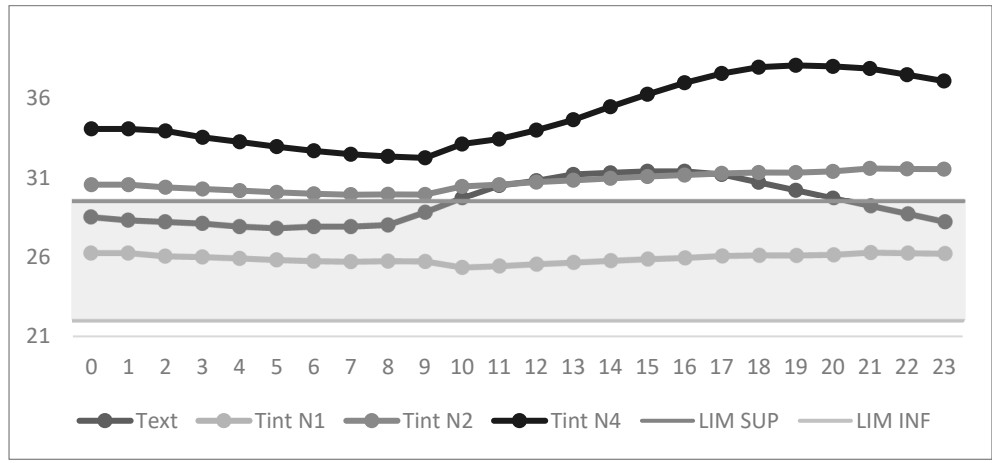

**Figure 12.** Outdoor and indoor temperatures on the hot day for case 1.

3.2.2. Case 2: Solar Protection + Natural Ventilation

The results showed that for the cool day in case 2, the building's indoor temperature (Tint) ranged between 25.9 °C and 27.6 °C, with a fluctuation of 1.7 °C and relative humidity levels of 65% (Figure 13). The mean radiant temperature (Tmrad) ranged between 26.4 °C and 27.8 °C, fluctuating similarly to the Tint. A general decrease in the building's temperatures was observed, with a reduction of 1.6 °C for Tint and 1.2 °C for Tmrad.

The temperature and relative humidity ranges at each level are observed in Table 10 by applying sun protection using awnings and allowing natural ventilation throughout the day. Compared to Case 0, the internal temperature fluctuations did not show significant changes, as in the previous case. Similarly, the maximum interior temperature decreased, with the middle floor (level 2) showing the most significant reduction in Tint, up to 2 °C, followed by the upper floor (level 4), with a decrease of 1.8 °C. The ground floor (level 1) showed the smallest reduction, with a decrease of 0.6 °C. Tmrad decreased similarly, reducing 0.8 °C at level 1, 1.7 °C at level 2, and 1.7 °C at level 4.

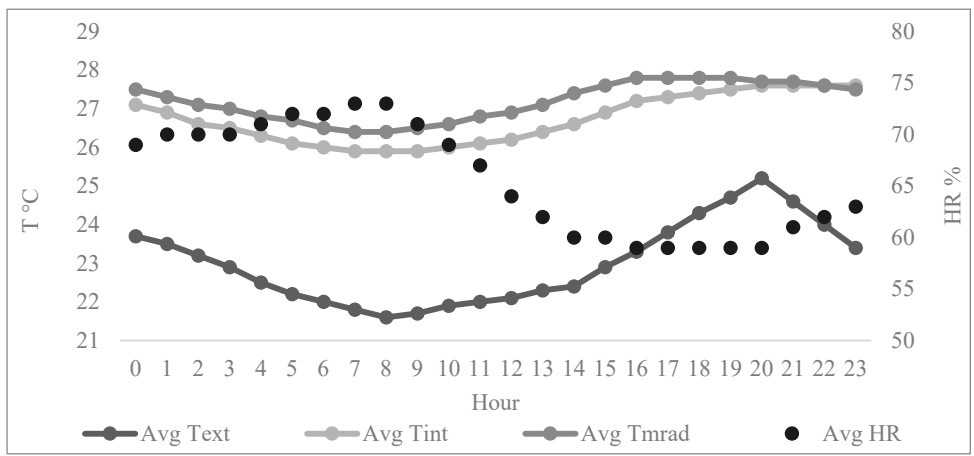

**Figure 13.** Average temperatures and relative humidity of the building on the cool day for case 2.

**Table 10.** Temperature range for the cool period for case 2.

| Environmental Variables | 1st Level (N1) | 2nd Level (N2) | 4th Level (N4) |
|---|---|---|---|
| Avg outside temperature | | 20.8–27.7 °C | |
| Avg Indoor temperature | 23.1–24.5 °C | 26.5–28 °C | 26.6–29.6 °C |
| Avg mean rad temperature | 23.1–24.1 °C | 27.1–28 °C | 27.2–30.4 °C |
| Avg relative humidity | 67–84% | 58–71% | 52–62% |

The PMV values in case 2 for the cool day were as follows: For level 1, a PMV of −0.3 indicated a neutral thermal sensation; for level 2, a PMV of 0.9 suggested a neutral sensation with a slight tendency toward warmth; and for level 4, a PMV of 1.1 indicated a slightly warm thermal sensation. These results reflect the improved comfort achieved through combined solar protection and natural ventilation.

For the warm day, the building's Tint ranged between 29.6 °C and 32.0 °C, with a fluctuation of 2.0 °C, combined with relative humidity levels of 76% (Figure 14). The Tmrad ranged between 30.1 °C and 32.1 °C. A slight overall decrease was observed, with Tint dropping by 0.8 °C and Tmrad by 0.6 °C. As in the cold-day scenario, internal temperature fluctuations did not show significant changes, as shown in Table 11. In this case, the best performance during this period was observed at the top floor (level 4), with a reduction of 1.2 °C in Tmrad, followed by the middle floor (level 2), with a decrease of 0.9 °C in Tint and 0.7 °C in Tmrad. The ground floor (level 1) did not experience any reduction during this period.

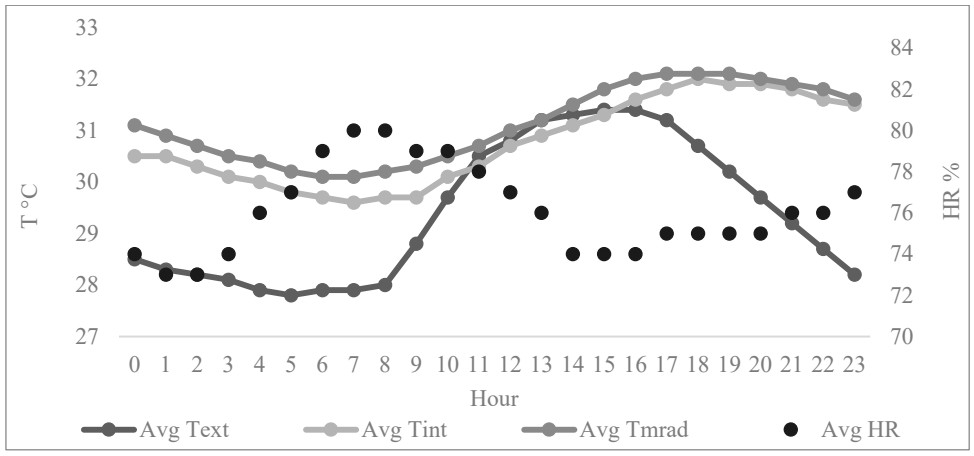

**Figure 14.** Average temperatures and relative humidity of the building on the hot day for case 2.

**Table 11.** Temperature range during the hot period for case 2.

| Environmental Variables | 1st Level (N1) | 2nd Level (N2) | 4th Level (N4) |
|---|---|---|---|
| Avg outside temperature | | 26–32.5 °C | |
| Avg Indoor temperature | 25.7–26.7 °C | 29.9–31.7 °C | 31.5–36.8 °C |
| Avg mean rad temperature | 25.4–26.3 °C | 30.4–31.6 °C | 32.4–37.5 °C |
| Avg relative humidity | 86–100% | 70–78% | 57–73% |

The PMV results can be seen in Table 12. On a warm day, the following PMV values were obtained: Level 1 had a PMV of 0.5, indicating a neutral thermal environment. Level 2 recorded a PMV of 2, which suggests a warm thermal sensation approaching hot. Finally, Level 4 showed a PMV of 3, indicating a hot environment.

**Table 12.** PMV for the cool and hot periods for case 2.

| | Cold Period | | | Hot Period | | |
|---|---|---|---|---|---|---|
| | 1st Level (N1) | 2nd Level (N2) | 4th Level (N4) | 1st Level (N1) | 2nd Level (N2) | 4th Level (N4) |
| Avg PMV | −0.4 | 0.6 | 0.9 | 0.5 | 2 | 3 |
| Avg PPD % | 17.9% | 9.7% | 18.4% | 15.3% | 63.1% | 92.1% |

When comparing the thermal performance results of the solar protection and natural ventilation strategy with the base case and assessing these against the comfort range of the adaptive model, we observe the following (Figures 15 and 16):

- On both cool and warm days, level 1 remained within the comfort range throughout the day, as in the previous scenario;
- On a cool day, the Tint on the middle level (N2) remained within the comfort range for 100% of the hours, consistent with the previous scenario.
- The top level (N4) achieved 100% of the hours within the comfort range, representing a significant 92% increase in comfort hours compared to the base case.

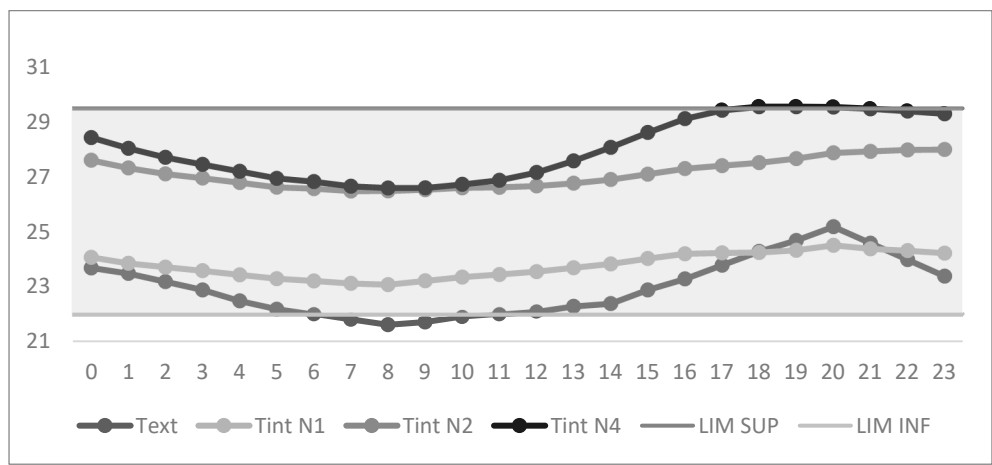

**Figure 15.** Range of comfort of the outdoor and indoor temperatures on the cool day for case 2.

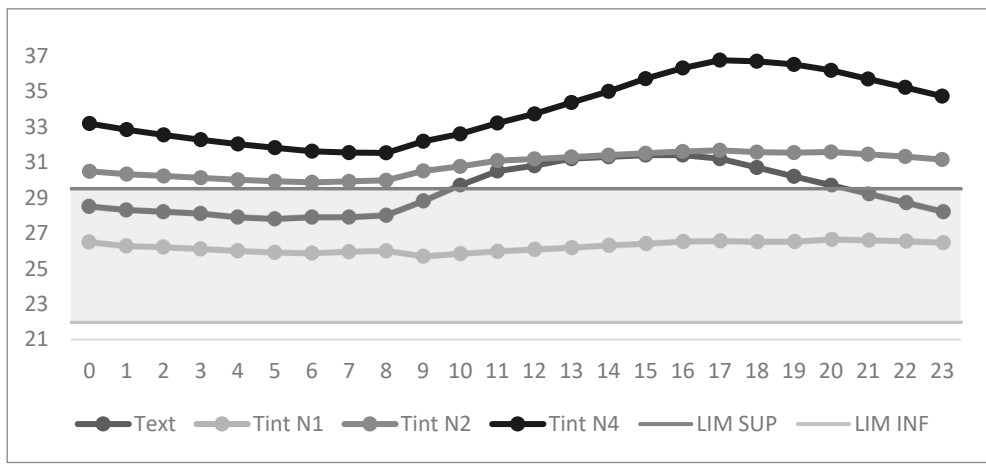

**Figure 16.** Range of comfort of the outdoor and indoor temperatures on the hot day for case 2.

3.2.3. Case 3: Solar Protection + Ventilation + Thermal Insulation

In case 3, results showed that for the cool day, the building's indoor temperature (Tint) ranged between 25.8 °C and 26.7 °C, with a fluctuation of 0.9 °C, combined with relative humidity levels of 68% (Figure 17). The mean radiant temperature (Tmrad) ranged between 26.3 °C and 26.8 °C, fluctuating similarly to the indoor temperature. Here, we observe a slight general reduction in the building's temperatures, with a decrease of 0.2 °C in Tint and 0.1 °C in Tmrad.

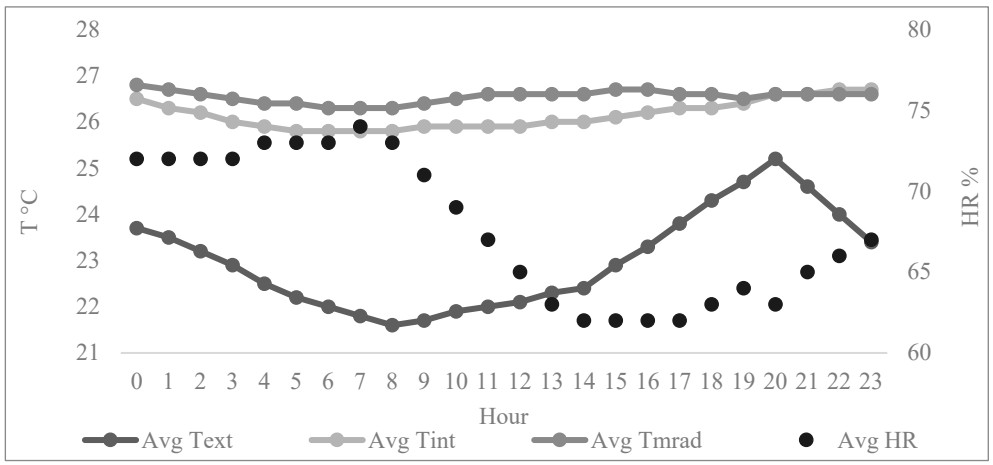

**Figure 17.** Average temperatures and relative humidity of the building on the cool day for case 3.

When solar protection through awnings, natural ventilation, and the addition of insulation to the facade and roof were applied, the ground floor (level 1) demonstrated the most significant reduction in indoor temperature (Tint), with a decrease of 0.8 °C.

Table 13 below shows the temperature and relative humidity ranges at each level. Compared to Case 0, internal temperature fluctuations show significant changes. In this case, the temperature fluctuations between minimum and maximum temperatures are smaller. The middle floor (level 2) showed a reduction of 3 °C. The upper floor (level 4) decreased its Tint by 2.7 °C, followed by the ground floor (level 1), which recorded a reduction of 2 °C. The mean radiant temperature (Tmrad) followed a similar trend, with a decrease of 2 °C at level 1, 2.7 °C at level 2, and 3.3 °C at level 4.

**Table 13.** Temperature range during the cool period for case 3.

| Environmental Variables | 1st Level (N1) | 2nd Level (N2) | 4th Level (N4) |
| --- | --- | --- | --- |
| Avg outside temperature | | 20.8–27.7 °C | |
| Avg Indoor temperature | 22.4–23.1 °C | 26.0–27.0 °C | 27.5–28.7 °C |
| Avg mean rad temperature | 22.3–22.9 °C | 26.5–27.0 °C | 28.3–28.8 °C |
| Avg relative humidity | 71–87% | 61–73% | 57–68% |

For the warm day, Figure 18, the building's Tint ranged between 28.4 °C and 29.3 °C, fluctuating 0.9 °C, and relative humidity levels were 82%. The Tmrad ranged between 28.5 °C and 29.1 °C. There was a general reduction in both Tint (1.4 °C) and Tmrad (1.6 °C) across the building.

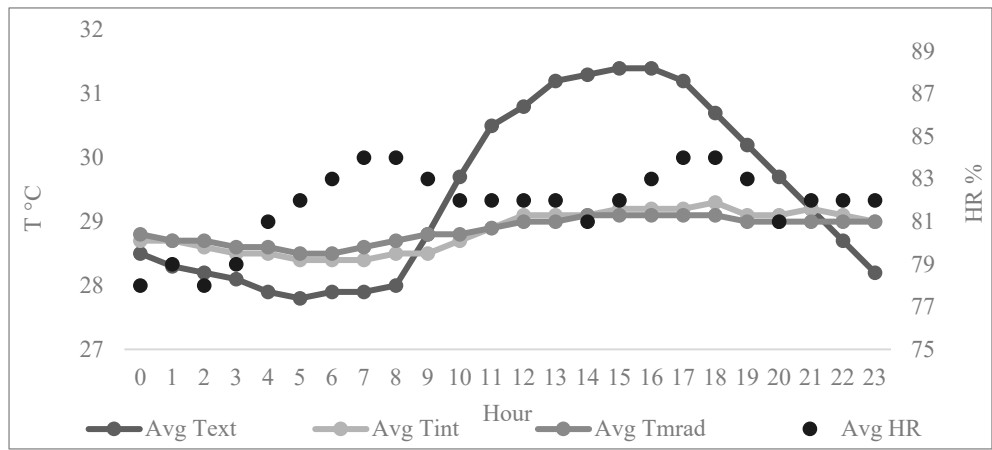

**Figure 18.** Average temperatures and relative humidity of the building on the hot day for case 3.

During this period, temperature fluctuations show slight changes (Table 14). In this case, the temperature fluctuations between minimums and maximums are more minor than in case 0. Similarly, the top floor (level 4) had the best performance, with a reduction of 5.9 °C in Tint and 6.5 °C in Tmrad, followed by the middle floor (level 2), which recorded a decrease of 3.1 °C in Tint and 3.1 °C in Tmrad. The ground floor (level 1) recorded the most minor reduction, with a decrease of 2 °C in Tint and 2.3 °C in Tmrad.

**Table 14.** Temperature range during the hot period for case 3.

| Environmental Variables | 1st Level (N1) | 2nd Level (N2) | 4th Level (N4) |
| --- | --- | --- | --- |
| Avg outside temperature | | 26–32.5 °C | |
| Avg Indoor temperature | 24.1–24.7 °C | 28.5–29.5 °C | 30.5–32.1 °C |
| Avg mean rad temperature | 23.7–24.0 °C | 28.6–29.2 °C | 31.3–32.2 °C |
| Avg relative humidity | 91–100% | 77–89% | 69–76% |

The PMV values for case 3 on a cool day were as follows (Table 15): For level 1, the PMV was −0.7, indicating a neutral thermal sensation with a slight tendency toward coolness; for level 2, the PMV was 0.4, reflecting a neutral indoor environment, which represents the ideal comfort scenario; and for level 4, the PMV was 0.8, indicating a neutral thermal sensation.

These results show that the combination of solar protection, natural ventilation, and insulation significantly improved thermal comfort, particularly on the top floor during the warm period, while also providing ideal comfort conditions on the middle level during the cool day.

**Table 15.** PMV for the cool and hot periods for case 3.

| | Cold Period | | | Hot Period | | |
|---|---|---|---|---|---|---|
| | 1st Level (N1) | 2nd Level (N2) | 4th Level (N4) | 1st Level (N1) | 2nd Level (N2) | 4th Level (N4) |
| Avg PMV | −0.7 | 0.4 | 0.8 | 0.1 | 1.9 | 2.1 |
| Avg PPD % | 25.9% | 8.6% | 13.2% | 15.9% | 38.3% | 71.6% |

For the warm day (Table 15), the PMV values were as follows: For level 1, the PMV was −0.2, indicating a neutral thermal sensation; for level 2, the PMV was 1.9, reflecting a slightly warm environment; and for level 4, the PMV was 2.1, indicating a warm thermal sensation.

When comparing the thermal performance results of the applied strategy (solar protection + natural ventilation + insulation) to the base case, in conjunction with the adaptive comfort model's range, we observe that, on a cool day, all levels maintained 100% of the hours within the comfort range, similar to case 2 (Figure 19).

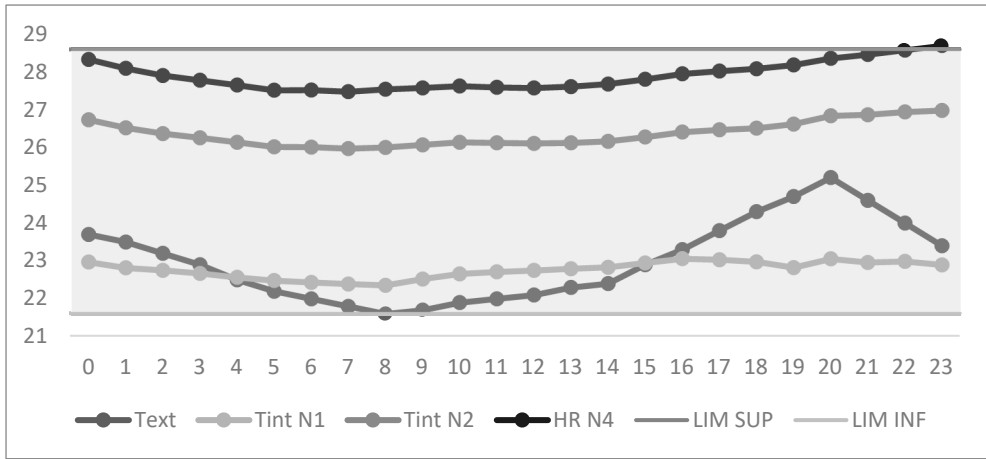

**Figure 19.** Outdoor and indoor temperatures on the cool day for case 3.

However, for the warm day (Figure 20), the middle floor (level 2) achieved 88% of the hours within the comfort range, a significant improvement compared to the base case (Ø). Despite these improvements, the top floor (level 4) remained in discomfort for 100% of the hours, highlighting the ongoing challenges in mitigating heat on higher floors, even with applying insulation, solar protection, and natural ventilation.

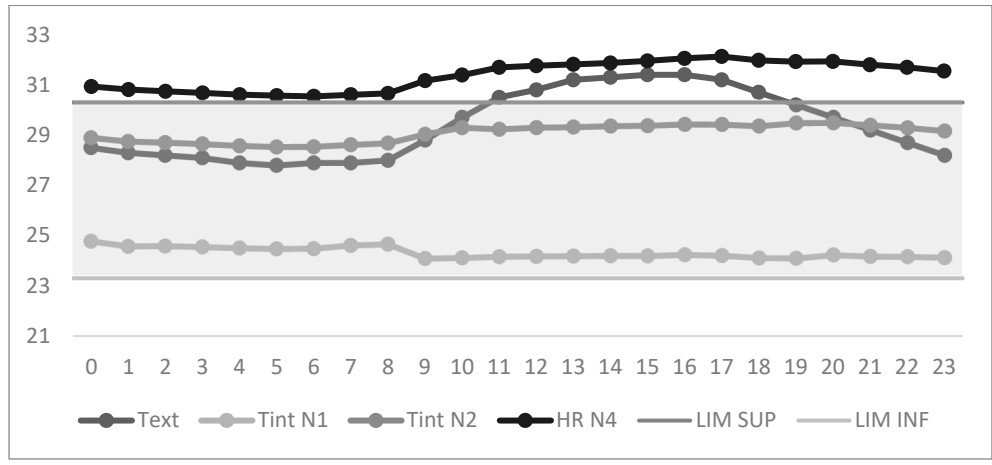

**Figure 20.** Outdoor and indoor temperatures on the hot day for case 3.

## 4. Discussion

### 4.1. Temperatures and Relative Humidity

Table 16 compares the base case with the different scenarios, meaning different passive strategies. As can be observed in these tables, the most efficient strategy to reduce Tint was scenario 3, which combined all strategies and showed the reduction according to the level and thermal zone of the building. In N1, corresponding to thermal zone 1, the maximum Tint was reduced by 2 °C; in N2, corresponding to thermal zone 2, it decreased by 3 °C; and in N4, corresponding to thermal zone 4, it was reduced by 2.7 °C.

**Table 16.** Comparative results for cases Ø, 1, 2, and 3.

| Case Studies | | Case Ø: Current Situation | Case 1: Solar Protection | Case 2: Solar Protection + Ventilation | Case 3: Solar Protection + Ventilation + Insulation |
|---|---|---|---|---|---|
| Cool day | Levels | Max Indoor temp °C | Max Indoor temp °C | Max Indoor temp °C | Max Indoor temp °C |
| | N1 | 25.1 °C | 24.4 °C | 24.5 °C | 23.1 °C |
| | N2 | 30.0 °C | 28.8 °C | 28.0 °C | 27 °C |
| | N4 | 31.4 °C | 31.3 °C | 29.6 °C | 28.7 °C |
| Hot day | Levels | Max Indoor temp °C | Max Indoor temp °C | Max Indoor temp °C | Max Indoor temp °C |
| | N1 | 26.7 °C | 26.3 °C | 26.7 °C | 24.7 °C |
| | N2 | 32.6 °C | 31.6 °C | 31.7 °C | 29.5 °C |
| | N4 | 38.0 °C | 38.1 °C | 36.8 °C | 32.1 °C |

On the selected warm day, it was observed that in N1 (thermal zone 1), the maximum Tint was reduced by 2 °C; in N2 (thermal zone 2), Tint decreased by 3.1 °C; and in N4 (thermal zone 4), Tint was reduced by 5.9 °C. Although both periods showed a significant reduction in Tint, the greatest reduction was noticeable on the hot day.

As for Tmrad fluctuated similarly to the interior temperature, with an average reduction of 2.4 °C on the cool day and 3 °C on the warm day. The relative humidity (HR) remained the same as the base case on the cool day at 64%, while on the hot day, it increased from 74% to 79%.

Scenario 2, where the passive strategy was to encourage the use of nighttime natural ventilation along with solar protection systems, also showed a reduction in Tint. In N1 and thermal zone 1, it decreased by 0.6 °C; in N2 and thermal zone 2, it decreased by 2 °C; and in N4, it reduced by 1.8 °C.

During the warm period, it was also observed that in N1, corresponding to thermal zone 1, Tint I did not present any variation; in N2 and thermal zone 2, it reduced by 0.9 °C; and in N4 and thermal zone 4, it decreased by 1.2 °C.

The period when the greatest temperature reduction was achieved with this strategy was on the cool day, due to lower external temperatures and lower relative humidity, allowing for better use of these conditions. On the other hand, higher humidity levels and elevated temperatures made the air warmer and increased the thermal sensation during the warm days.

As for Tmrad, it fluctuated similarly to the interior temperature, with an average reduction of 2.3 °C on the cool day and 1.3 °C on the warm day. Relative humidity (HR) on the cool day decreased from 64% to 63%, while on the warm day, it increased from 74% to 78%.

The scenario with the lowest temperature reduction was Case 1, where solar protection for windows was implemented. It was observed that in N1 (thermal zone 1), the temperature decreased by 0.7 °C; in N2 (thermal zone 2), it dropped by 1.2 °C; and in N4 (thermal zone 4), it decreased by 0.1 °C.

During the warm period, the results showed that in N1 (thermal zone 1), the Tint dropped by 0.4 °C; in N2 (thermal zone 2), the Tint decreased by 1 °C; and in N4 (thermal zone 4), it increased by −0.1 °C.

The mean radiant temperature (Tmrad) fluctuated similarly to the indoor temperature, with an average reduction of 1.1 °C on the cool day and 1.7 °C on the warm day. Relative humidity (HR) decreased from 64% to 63% on the cool day, while on the warm day, it increased from 74% to 78%.

There was a slight reduction in Tint in both periods, with the most significant improvements seen on the cool day. This is mainly due to the sun's trajectory, which is primarily vertical, making N4 the level with the worst performance.

It is important to consider that scenarios 2 and 3 should be taken into account during the initial design phase of the building. In strategy 2, window placement must be optimized to enhance natural ventilation throughout the spaces.

*4.2. Comfort Models*

According to the Fanger model, primarily used to evaluate mechanically conditioned environments, the thermal sensation index ranges from 0 to +3 and 0 to −3, where positive values indicate warmer temperatures. These results are shown in Table 17. On the cool day, level 1 maintained neutral values with ranges between 0.1 and −0.3 in all cases. At level 2, comfort index values were reduced in case 2 from warm to slightly warm, with ranges between 2 and 1, and in case 3, the sensation was neutral. Level 4 remained warm (2) in cases 1 and 2, while in case 3, the sensation reduced to slightly warm (1).

On a warm day, level 1 maintained neutral values, ranging between 0.5 and −0.4 in all cases. At level 2, comfort index values were reduced to 3 from warm to slightly warm, with ranges between 2 and 1. At level 4, the sensation remained hot (2) in cases 1 and 2, while in case 3, it reduced to warm (1).

With the adaptive model, shown in Table 18, the lower level (N1) and the middle level (N2) met the 80% acceptability limits on the cool day in all cases. The upper level (N4) met the 80% acceptability limits in cases 2 and 3. On a warm day, level 1 met the 80% acceptability limits in all cases, level 2 met these limits in case 3, while level 4 did not reach the comfort ranges. This adaptive model is more suitable for naturally ventilated buildings, as recommended by ASHRAE Standard 55.

As observed in both models, on a cool day, the lower level (N1) was within the comfort range in all studied cases, including the base model. The middle level (N2) in the static model reached comfort in case 3, while in the adaptive model, it was within the comfort range in all implemented strategies. The upper level (N4), although transitioning from a warm sensation to slightly warm, did not achieve comfortable values in the static model. However, with the adaptive model, it achieved comfortable temperature ranges with strategies 2 and 3.

On the warm day, the lower level (N1) remained comfortable in all studied cases, including the base model. In the static model, the middle level (N2) shifted from a warm to a slightly warm sensation in case 3 but did not reach comfortable values with any of the implemented strategies. In contrast, the adaptive model achieved comfort in case 3. The upper level (N4), while transitioning from hot to warm, did not reach comfortable values in the static model, and the adaptive model also failed to bring it into comfortable temperature ranges in any of the cases. Although the temperature was significantly reduced, it was not enough to reach comfort.

**Table 17.** Scale for the thermal sensation PMV on cool and hot days.

| Range Comfort Scale | Study Cases | Case Ø: Current Situation | | | Case 1: Current Situation + Solar Protection | | | Case 2: Current Situation + Solar Protection + Ventilation | | | Case 2: Current Situation + Solar Protection + Ventilation + Insulation | | |
|---|---|---|---|---|---|---|---|---|---|---|---|---|---|
| | | Levels | PMV | Scale | Levels | PMV | Scale | Levels | PMV | Scale | Levels | PMV | Scale |
| 3 Hot | Cool day | N1 | −0.1 | Neutral | N1 | −0.4 | Neutral | N1 | −0.4 | Neutral | N1 | −0.7 | Neutral |
| 2 Warm | | N2 | 1.4 | slightly warm | N2 | 1.0 | slightly warm | N2 | 0.6 | slightly warm | N2 | 0.4 | Neutral |
| 1 Slightly warm | | N4 | 1.9 | warm | N4 | 1.4 | slightly warm | N4 | 1.0 | warm | N4 | 0.8 | slightly warm |
| 0 Neutral | Hot day | N1 | 0.4 | Neutral | N1 | 0.4 | Neutral | N1 | 0.5 | Neutral | N1 | −0.2 | Neutral |
| −1 Slightly cool | | N2 | 2.2 | warm | N2 | 1.9 | warm | N2 | 1.9 | warm | N2 | 1.4 | slightly warm |
| −2 cool / −3 cold | | N4 | 3.4 | hot | N4 | 3.4 | hot | N4 | 2.5 | hot | N4 | 2.1 | warm |

**Table 18.** Scale for the comfort adaptive model on cool and hot days.

| Study Cases | Case Ø: Current Situation | | | Case 1: Current Situation + Solar Protection | | | Case 2: Current Situation + Solar Protection + Ventilation | | | Case 2: Current Situation + Solar Protection + Ventilation + Insulation | | |
|---|---|---|---|---|---|---|---|---|---|---|---|---|
| | Levels | Max Indoor Temp °C | Scale | Levels | Max Indoor Temp °C | Scale | Levels | Max Indoor Temp °C | Scale | Levels | Max Indoor Temp °C | Scale |
| Cool Day | N1 | 25.1 °C | ✔ | N1 | 24.4 °C | ✔ | N1 | 24.5 °C | ✔ | N1 | 23.1 °C | ✔ |
| | N2 | 30.0 °C | ✗ | N2 | 28.8 °C | ✔ | N2 | 28.0 °C | ✔ | N2 | 27.0 °C | ✔ |
| | N4 | 31.4 °C | ✗ | N4 | 31.3 °C | ✗ | N4 | 29.6 °C | ✔ | N4 | 28.7 °C | ✔ |
| Hot Day | N1 | 26.7 °C | ✔ | N1 | 26.3 °C | ✔ | N1 | 26.7 °C | ✔ | N1 | 24.7 °C | ✔ |
| | N2 | 32.6 °C | ✗ | N2 | 31.6 °C | ✗ | N2 | 31.7 °C | ✗ | N2 | 29.5 °C | ✔ |
| | N4 | 38.0 °C | ✗ | N4 | 38.1 °C | ✗ | N4 | 36.8 °C | ✗ | N4 | 32.1 °C | ✗ |

## 5. Conclusions

The main objective of this study was to evaluate passive strategies for a hot and humid climate to reduce thermal discomfort in a social housing unit in the Dominican Republic. Three scenarios were established: one with solar protection, another implementing natural ventilation, and the third adding thermal insulation to the facade and roof to allow for a comparison with the current model. The thermal sensation index was analyzed using the PMV-PPD and the adaptive model. Natural ventilation, orientation, and shading are critical factors in design strategies to improve the thermal performance of buildings passively. After finishing the study, the following conclusions were reached:

- The results demonstrated that these social housing units reached temperatures outside the comfort levels, exceeding the comfort ranges established by the static and adaptive models.
- When evaluating the passive strategies, it was confirmed that their implementation in housing could lead to significant temperature reductions, especially in the third scenario, where a combination of all the evaluated measures was applied.
- In scenario 1, where solar protection was implemented, the average indoor temperature improved by 0.8 °C on the cool day and 0.6 °C on the warm day. In scenario 2, with the addition of nighttime ventilation, the improvement was 1.9 °C on the cool day and 0.9 °C on the warm day. In scenario 3, with insulation added, the improvement was 2.4 °C on the cool day and 2.9 °C on the warm day. These temperature improvements bring the interior comfort levels closer to the acceptable ranges in static and adaptive models.
- The differences in comfort depending on the floor showed to have relevance in comfort. That emphasizes the need for solar protection, not only in windows but even in roofs, to avoid solar gains on the top floors.
- The passive strategies are not enough to ensure adequate comfort levels. For instance, regarding relative humidity, it was noted that the implementations greatly helped maintain the humidity percentage in both periods. This can lead to high humidity levels, confirming that passive strategies alone are insufficient for this type of climate. Mechanical means are necessary to achieve thermal comfort, especially during warmer periods.
- The results suggest that high levels of discomfort are, to some extent, in addition to the lack of HVAC systems (heating, ventilation, and air conditioning), caused by the building's inadequate climate adaptation.

As a limitation of the study, the analysis of the costs associated with each intervention is out of the scope. Given the variability in prices and the current economic situation,

examining the costs of the different interventions is an important topic that warrants a separate study.

**Author Contributions:** Conceptualization, D.A.-M.; methodology, D.A.-M. and A.Q.-G.; software, D.A.-M.; validation, I.G.-G., A.Q.-G. and F.A.M.; formal analysis, D.A.-M.; investigation, D.A.-M.; resources, D.A.-M.; data curation, A.Q.-G. and F.A.M.; writing—original draft preparation, D.A.-M.; writing—review and editing, A.Q.-G.; visualization, F.A.M.; supervision, I.G.-G. and A.Q.-G.; project administration, I.G.-G.; funding acquisition, I.G.-G. All authors have read and agreed to the published version of the manuscript.

**Funding:** Authors A.Q.-G. and I.G.-G. gratefully thank the Vicerrectorado de Investigación of the Polytechnic University of Valencia for funding the project "Parametrización de impactos sociales en vivienda: Análisis de Ciclo de Vida Social" (PAID-06-23).

**Institutional Review Board Statement:** Not applicable.

**Informed Consent Statement:** Not applicable.

**Data Availability Statement:** Data are available from the sources described in the article. Additional material can be provided upon demand by writing to the corresponding author.

**Conflicts of Interest:** The authors declare no conflicts of interest. The funders had no role in the design of the study, in the collection, analyses, or interpretation of data, in the writing of the manuscript, or in the decision to publish the results.

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
