# Peer review of "Evaluation of Passive Strategies for Achieving Hygrothermal Comfort in Social Housing Buildings in the Dominican Republic"

_sustainability, doi:10.3390/su17083416_

Round 1

Reviewer 1 Report

Comments and Suggestions for Authors

This research analyzed the thermal behavior of typical social housing buildings through energy simulation, aiming to emphasize the importance of passive strategies in improving comfort without using air conditioning in warm and humid climates. The climate characteristics of the Dominican Republic were the main analysis object. The simulation was carried out using Open Studio, which uses the Energy Plus calculation engine. Three case studies implementing passive measures were established to achieve temperatures within the comfort range of the housing prototype. The results show that under warm and humid climate conditions, the combined use of passive strategies can significantly reduce the temperature, by 2.8°C in cooler periods and 3.2°C in warmer periods. Suggestions for Improvement: 1. The author designed three schemes for the passive strategies of the building and obtained the indoor environmental parameter data and the thermal evaluation value index PMV - PPD value through the model. In the second part of the paper, each scheme was compared and analyzed with the basic scheme respectively. It is recommended to directly summarize and compare the data of the three schemes with those of the basic scheme to make the data comparison more intuitive and reduce the number of charts. 2. There are many repetitions of table header numbers and figure title numbers in the paper, such as in lines 291, 305, 306, 316, etc. It is recommended to rearrange the numbers to avoid errors. 3. There are many cases in the paper where the figure titles do not match the actual data display, such as in lines 241, 215, 270, 279, etc. Please check whether there are duplications or errors.

  1. In the paper, there are cases where the table data do not match the text description, or the data of N1, N2 and N4 are obviously the same, such as in lines 247, 298, 306, 359, 371, 415, 427, 498, 521, 527, etc. Please check whether there are duplications or errors.5. The PMV - PPD data are compared in the paper. It is recommended to supplement the hourly data comparison. Why is the fluctuation of PMV value in Table 13 not large when the temperatures and humidities of N2 and N4 are inconsistent on a hot day?

  1. 6. In the paper, for the physiological variables, the average value of clothing insulation performance is 0.5 (grams per degree Celsius) and the metabolic rate is 1.2 (metabolic equivalents). Why are the values the same for hot days and cool days?
Comments on the Quality of English Language

 The English could be improved to more clearly express the research.

Author Response

First of all, we want to thank the reviewers for the immense amount of work they put in to improve our paper, it has been tremendously helpful to go through all the comments.

Reviewer 1

This research analyzed the thermal behavior of typical social housing buildings through energy simulation, aiming to emphasize the importance of passive strategies in improving comfort without using air conditioning in warm and humid climates. The climate characteristics of the Dominican Republic were the main analysis object. The simulation was carried out using Open Studio, which uses the Energy Plus calculation engine. Three case studies implementing passive measures were established to achieve temperatures within the comfort range of the housing prototype. The results show that under warm and humid climate conditions, the combined use of passive strategies can significantly reduce the temperature, by 2.8°C in cooler periods and 3.2°C in warmer periods. 

Suggestions for Improvement: 

  1. The author designed three schemes for the passive strategies of the building and obtained the indoor environmental parameter data and the thermal evaluation value index PMV - PPD value through the model. In the second part of the paper, each scheme was compared and analyzed with the basic scheme respectively. It is recommended to directly summarize and compare the data of the three schemes with those of the basic scheme to make the data comparison more intuitive and reduce the number of charts. 

Thank you so much for the advice. We have merged the PMV tables in the results section. We have also merged the three tables in Section 4 into one (Table 16). Now, the text is easier to follow.

  1. There are many repetitions of table header numbers and figure title numbers in the paper, such as in lines 291, 305, 306, 316, etc. It is recommended to rearrange the numbers to avoid errors. 

We are sorry for all the repetitions and errors. We have reviewed carefully the manuscript to correct those errors.

  1. There are many cases in the paper where the figure titles do not match the actual data display, such as in lines 241, 215, 270, 279, etc. Please check whether there are duplications or errors.

Thank you so much for carefully checking the figures. We have corrected our mistakes.

  1. In the paper, there are cases where the table data do not match the text description, or the data of N1, N2 and N4 are obviously the same, such as in lines 247, 298, 306, 359, 371, 415, 427, 498, 521, 527, etc. Please check whether there are duplications or errors.

Thank you so much, it was an error produced while adapting the manuscript to the format of the journal. We have corrected the following mistakes: Table3, Table 4, Table 7

  1. The PMV - PPD data are compared in the paper. It is recommended to supplement the hourly data comparison. Why is the fluctuation of PMV value in Table 13 not large when the temperatures and humidities of N2 and N4 are inconsistent on a hot day?

We have corrected the results of the PMV values, as there were some mistakes during the adaptation process of the manuscript to the journal template. Now the results have been rechecked and corrected.

We have not added the hourly PMV, as this information would increase considerably the length of the paper and would contradict the idea of most reviewers of reducing the amount of figures and tables.

  1. In the paper, for the physiological variables, the average value of clothing insulation performance is 0.5 (grams per degree Celsius) and the metabolic rate is 1.2 (metabolic equivalents). Why are the values the same for hot days and cool days?

We have added the following text in section 2.2 to explain this decision:

“The clothing insulation performance is considered the same for the cool and the hot period. The reason for this is the low temperature differences during the year. This has been confirmed through informal interviews with residents of Santo Domingo.”

Reviewer 2 Report

Comments and Suggestions for Authors

Passive strategies for achieving hygrothermal comfort in social housing buildings was investigated through energy simulation in this study. There are some comments:

  1. The introduction is insufficient, it should be added literature to explain the progress of hygrothermal comfort study. Especially focus on passive strategies.
  2. The figures in “Materials and Methods” like Figure 1 and Figure 5 is blurring, in my view, the authors should use with picture high resolution.
  3. Figure 1, Figure 3, Figure 4, Figure 6…… does not explain in text, what is its function?
  4. Energy simulation model should be verified. And the temperature and humidity measuring method should describe.
  5. There are too much tables in this study, the authors should merge some tables, for example “Table 8” and “Table 9”.
  6. In conclusions “The primary reason for the high levels of thermal discomfort in these social housing units is not the lack of economic resources but rather the lack of innovation and the application of bioclimatic knowledge”, this conclusion cannot be got from this investigation.
  7. Three scenarios: “The base model + solar protection”; “The base model + solar protection + natural ventilation”; and “The base model + solar protection + natural ventilation + insulation” were used for achieving hygrothermal comfort, however, in conclusions, authors said that: ”It has been shown that building orientation and layout are crucial in reducing energy consumption in both cold and hot climates”. In my view, the effect of passive strategies on temperature and humidity should discuss clear.
  8. The conclusion does not match the research content.
Comments on the Quality of English Language

 The English could be improved to more clearly express the research

Author Response

First of all, we want to thank the reviewers for the immense amount of work they put in to improve our paper, it has been tremendously helpful to go through all the comment  

Reviewer 2

Passive strategies for achieving hygrothermal comfort in social housing buildings was investigated through energy simulation in this study. There are some comments:

  1. The introduction is insufficient, it should be added literature to explain the progress of hygrothermal comfort study. Especially focus on passive strategies.

We have improved the introduction and added several references to explain the progress of comfort studies using passive strategies.

  1. The figures in “Materials and Methods” like Figure 1 and Figure 5 is blurring, in my view, the authors should use with picture high resolution.

Thank you for pointing it out. We have improved the quality of the figures.

  1. Figure 1, Figure 3, Figure 4, Figure 6…… does not explain in text, what is its function?

We have added the following piece of text to go along Figure 1:

“The main idea is to analyze the case study in four different situations. The first one should reflect the current situation of most social housing projects in Santo Domingo, which, as it is explained in following sections, lacks any kind of climate adaptation strategy. The second situation includes the addition of solar protection. The third strategy adds utilizing natural ventilation strategies to avoid overheating. The last situation that must be studied is adding thermal insulation, which should be combined with the rest of the strategies. These four situations in which the case study is analyzed are depicted in Figure 1.”

We have added the following text for Figures 2 and 3:

“Figures 2 and 3 show the architectural floor plan of the building and the front elevation.”

We have removed Figure 4, as can be considered redundant.

Overall, all the figures have now at least a sentence to go along with them.

  1. Energy simulation model should be verified. And the temperature and humidity measuring method should describe.

We have added an explanation in section 2.3 the clarify the reason behind not doing in situ measurements:

“Due to the difficulty of taking and using expensive measuring equipment in the area, the paper is based purely on simulations. In situ measurements are outside the scope of this study.”

  1. There are too many tables in this study, the authors should merge some tables, for example “Table 8” and “Table 9”.

We have merged several tables to reduce the total amount.

  1. In conclusions “The primary reason for the high levels of thermal discomfort in these social housing units is not the lack of economic resources but rather the lack of innovation and the application of bioclimatic knowledge”, this conclusion cannot be got from this investigation.

You are completely right. The economic part is outside the scope of this article. We have reworked the sentence to fit our results (Section 5):

“Overall, the results suggest that high levels of discomfort are, to some extent, caused by the building's inadequate climate adaptation.”

  1. Three scenarios: “The base model + solar protection”; “The base model + solar protection + natural ventilation”; and “The base model + solar protection + natural ventilation + insulation” were used for achieving hygrothermal comfort, however, in conclusions, authors said that: ”It has been shown that building orientation and layout are crucial in reducing energy consumption in both cold and hot climates”. In my view, the effect of passive strategies on temperature and humidity should discuss clear.

Sorry for the inconvenience. That sentence has been removed, as it did not reflect the results obtained.

“Para la orientación prevalente las medidas de adaptación climática demuestran una mejora del confort en el interior”

  1. The conclusion does not match the research content.

We have reformulated the conclusions to better reflect the content of the research.

Reviewer 3 Report

Comments and Suggestions for Authors

Overall, the study is of high quality, logically structured, easy for readers to follow, but quite lengthy. The tables and their presentation follow one another in an identical structure, which does not help to maintain the reader's attention.
My conclusions follow the structure of the paper:

1 Introduction - 2. Materials and Methods
I think it is important to underline how the study carried out by the authors reflects a gap in knowledge and goes beyond the work of Fanger (1973), Givoni (1969) and Olgyay (1998) on which they are based. The vertical differentiation observed in terms of thermal comfort, the way this is captured by simulations, and the differential effect of different interventions (solar protection/+ natural ventilation/+insulation) are definitely important results - and the authors should emphasise this in the study. 

3. Results section
I suggest that the authors should consider showing the impact of the passive strategies used (solar protection/+ natural ventilation/+insulation) on the evolution of the indicators studied in relation to the baseline model on cool and warm days. This would allow to present the results in just two larger tables and to better show the impact of the strategies used.

4. Discussion
It would help the social benefits of the results, and perhaps also increase policy attention, if the authors at least gave a rough estimate of the expenditure per building that would be required to implemantation of passive strategies.

Minor comments
- It is not clear why the third floor was left out of the simulation - an explanation is needed
- The handling of references is not consistent - please follow the reference scheme expected by the journal! (line 177)
- In line 35 "Later, in 1973, P. O. Fanger [4] published his method for evaluating thermal" - this item is not included in the bibliography. Need to check references!!!!
- Spanish language is sometimes used in the paper 
          - Table 1 - in column header (Conductividad térmica)
          - variable name (Façade concrete block)

Author Response

First of all, we want to thank the reviewers for the immense amount of work they put in to improve our paper, it has been tremendously helpful to go through all the comments.

Reviewer 3

Overall, the study is of high quality, logically structured, easy for readers to follow, but quite lengthy. The tables and their presentation follow one another in an identical structure, which does not help to maintain the reader's attention.
My conclusions follow the structure of the paper:

1 Introduction -2. Materials and Methods
I think it is important to underline how the study carried out by the authors reflects a gap in knowledge and goes beyond the work of Fanger (1973), Givoni (1969) and Olgyay (1998) on which they are based. The vertical differentiation observed in terms of thermal comfort, the way this is captured by simulations, and the differential effect of different interventions (solar protection/+ natural ventilation/+insulation) are definitely important results - and the authors should emphasise this in the study. 

Thank you for pointing it out. We have added the following paragraph:

“Also, the differences in comfort depending on the floor under study are relevant to emphasize the need for solar protection, not only in windows but even in roofs, to avoid solar gains on the top floors.”

  1. Results section
    I suggest that the authors should consider showing the impact of the passive strategies used (solar protection/+ natural ventilation/+insulation) on the evolution of the indicators studied in relation to the baseline model on cool and warm days. This would allow to present the results in just two larger tables and to better show the impact of the strategies used.

We have reduced the amount of tables to facilitate the reading and understanding of the results. Now the results are presented in larger tables comparing the different cases.

  1. Discussion
    It would help the social benefits of the results, and perhaps also increase policy attention, if the authors at least gave a rough estimate of the expenditure per building that would be required to implemantation of passive strategies.

Estimating the prices is outside the scope of this article. Due to the variability in prices and considering the current economic situation, assuming the cost of the different interventions and its variation along the years would constitute a paper on its own. We have added the following paragraph to account for this:

“The analysis of the costs associated with each intervention is not included in this study. Given the variability in prices and the current economic situation, examining the costs of the different interventions is an important topic that warrants a separate study.”

Minor comments
- It is not clear why the third floor was left out of the simulation - an explanation is needed

We have not included the third floor due to the fact that levels two and three can be considered equivalent, as neither of them is in contact with the floor nor the roof. We have used this simplification due to the fact that the main point of the study is to analyze how the different passive strategies affect the building, not obtaining the overall results of the apartment building. Adding the third floor would add even more figures and tables to the manuscript.

We have added the following paragraph to the methodology section to clarify the reason behind our choice:

“The floors included in the study are the first (N1), second (N2), and fourth (N4). The third floor was not included because it can be considered equivalent to N2, as neither is in contact with the floor or the roof.”

- The handling of references is not consistent - please follow the reference scheme expected by the journal! (line 177)

Sorry for the inconvenience. We have corrected the mistake.
- In line 35 "Later, in 1973, P. O. Fanger [4] published his method for evaluating thermal" - this item is not included in the bibliography. Need to check references!!!!

We apologize. We have included the reference and checked the rest of them.
- Spanish language is sometimes used in the paper 
          - Table 1 - in column header (Conductividad térmica)

          - variable name (Façade concrete block)

We are extremely sorry for the mistake and thank you for pointing it out. We have gone through the document and replaced the Spanish with the appropriate English word. We have maintained the word façade, as it is a correct word in the English language.

Round 2

Reviewer 1 Report

Comments and Suggestions for Authors

no more. It is finished to revive that problems and suggestions are given to author by ourselves.

Reviewer 2 Report

Comments and Suggestions for Authors

In my view, I believe the article can be accepted in this form